# Non-equilibrium crystallization pathways of manganese oxides in aqueous solution

Wenhao Sun[1], Daniil A. Kitchaev [2], Denis Kramer [3] & Gerbrand Ceder[1,2,4]

Aqueous precipitation of transition metal oxides often proceeds through non-equilibrium phases, whose appearance cannot be anticipated from traditional phase diagrams. Without a precise understanding of which metastable phases form, or their lifetimes, targeted synthesis of specific metal oxides can become a trial-and-error process. Here, we construct a theoretical framework to reveal the nanoscale and metastable energy landscapes of Pourbaix ($E$-$p$H) diagrams, providing quantitative insights into the size–dependent thermodynamics of metastable oxide nucleation and growth in water. By combining this framework with classical nucleation theory, we interrogate how solution conditions influence the multistage oxidation pathways of manganese oxides. We calculate that even within the same stability region of a Pourbaix diagram, subtle variations in pH and redox potential can redirect a non-equilibrium crystallization pathway through different metastable intermediates. Our theoretical framework offers a predictive platform to navigate through the thermodynamic and kinetic energy landscape towards the rational synthesis of target materials.

[1] Materials Sciences Division, Lawrence Berkeley National Laboratory, Berkeley, CA 94720, USA. [2] Department of Materials Science and Engineering, Massachusetts Institute of Technology, Cambridge, MA 02139, USA. [3] Engineering Sciences, University of Southampton, Southampton SO17 1BJ, UK. [4] Department of Materials Science and Engineering, University of California, Berkeley, CA 94720, USA. Correspondence and requests for materials should be addressed to W.S. (email: wenhaosun@lbl.gov) or to G.C. (email: gceder@berkeley.edu)

Transition metal oxides drive the functionality of an enormous range of technological materials; spanning battery cathodes, catalysts, fuel cells, magnetic media, and more. The breadth of transition metal oxide applications largely stems from the diversity of their electronic, optical, and magnetic properties, which can be tuned as a function of the crystal structure and metal oxidation state[1]. Understanding how to rationally synthesize metal oxides in desired phases, with desired oxidation states, is central towards unlocking the full potential of transition metal oxide design. The manganese oxides are a remarkable example of structural and oxidation-state diversity, spanning more than 30 phases over oxidation states from $Mn^{2+}$ to $Mn^{7+}$[2]. This broad structural diversity makes manganese oxides relevant for a variety of applications; for example, the spinel λ-$MnO_2$ phase is an important lithium-ion battery cathode[3]; ramsdellite-$MnO_2$ is used in alkaline batteries[4]; and $Mn^{3+}$ containing phases, such as Hausmannite $Mn_3O_4$ and bixbyite $Mn_2O_3$, are precursors to water-splitting catalysts[5–8]. The precipitation and dissolution of various manganese (oxy)hydroxides also play important roles in redox-active biogeochemical processes, such as mediating oceanic $O_2/H_2S$ cycles[9], microbial metabolic cycles[10], and soil chemistry[11]. Unfortunately, the structural diversity of the manganese oxides also results in a myriad of possible crystallization pathways in solution, which often leads to poor phase-control during crystal growth. Although synthesis recipes to specific manganese oxide phases have been identified and cataloged[5,12], a comprehensive understanding of the thermodynamic and kinetic processes that drive phase-selection during aqueous crystallization remains elusive.

Previously, we found that spectator ions can be important $MnO_2$ structure-directing agents, as intercalation of aqueous alkali cations such as $Li^+$, $Na^+$, $K^+$, etc. can stabilize the metastable α-$MnO_2$, δ-$MnO_2$, and λ-$MnO_2$ polymorph frameworks at off-stoichiometric compositions[13]. However, precipitation of manganese oxides often proceeds by Ostwald's 'Rule of Stages'[14], where a variety of metastable manganese oxides and oxyhydroxides nucleate and grow prior to the formation of the equilibrium phase[15,16]. These non-equilibrium crystallization pathways can occur both with[17,18], and without[19] impurity ions in solution. Further complicating the matter, small variations in precursor choice and solution redox conditions can change which metastable phases are observed, as well as their lifetimes, even when the final equilibrium product remains unchanged[20]. Understanding how solution chemistry influences structure-selection along a non-equilibrium crystallization pathway would enable the rational design of aqueous synthesis routes; either towards desirable metastable phases, or away from long-lived metastable byproducts and towards the synthesis of a desired equilibrium phase[21,22]. Importantly, a predictive understanding of hydrothermal synthesis developed on the manganese oxides could be broadly generalized and applied to other transition metal oxide systems.

According to classical nucleation theory, a metastable phase can precipitate first from a supersaturated solution if it has a lower nucleation barrier than the stable phase[23–25], The nucleation barrier takes the form:

$$\Delta G_c \propto \frac{(\eta\gamma)^3}{(-RT\ln\sigma)^2} \quad (1)$$

where $\eta\gamma$ is the shape-averaged surface energy, $\Delta G_{Bulk} = -RT\ln\sigma$ is the bulk thermodynamic driving force for crystallization, and σ is the supersaturation. A metastable phase exhibits a smaller $\Delta G_{Bulk}$ for its formation than the stable phase, but the metastable phase can still dominate the kinetics of nucleation if this smaller driving force is further compensated by a lower surface energy[26–28]. Calorimetry experiments have shown that many bulk metastable polymorphs have lower surface energies than their corresponding stable phases[29], and that surface energy differences between divalent (MO), trivalent ($M_2O_3$), and spinel ($M_3O_4$) metal oxides can shift redox equilibria at the nanoscale by orders of magnitude in oxygen fugacity[30]. Incorporating the influence of surface energies on solid-aqueous equilibria at the nanoscale is therefore central towards rationalizing the non-equilibrium crystallization pathways of transition metal oxides in water.

In this work, we extend $E$-pH diagrams, also known as Pourbaix diagrams, to capture the size-dependent thermodynamics of metastable oxide nucleation and growth. First, we construct a thermodynamic grand potential for a metal oxide being acted upon by an external water reservoir with given pH and redox potential, which provides a free-energy axis to Pourbaix diagrams. This free-energy axis is then generalized to incorporate surface energies, enabling the construction of size-dependent Pourbaix diagrams, which capture how particle size influences solid-aqueous equilibria at the nanoscale—where nucleation initiates. The Pourbaix free-energy axis also visualizes how changes in $E$ and pH shift the metastable energy landscape, altering the bulk thermodynamic driving forces between reactant and product phases. By combining the Pourbaix potential with classical nucleation theory, we show that even when crystallization starts from the same precursor and ends with the same equilibrium phase, minor variations in $E$ and pH can qualitatively change which metastable phases form on the crystallization pathway. Our theoretical framework offers a predictive platform to map how experimental parameters influence non-equilibrium crystallization pathways in redox-active systems, and represents an important step towards a predictive theory of materials synthesis.

## Results

**A thermodynamic grand potential for Pourbaix diagrams.** Traditionally, Pourbaix diagrams are constructed by using the Nernst equation to calculate $E$-pH boundaries between aqueous phase stability regions[31]. However, Pourbaix diagrams constructed by this approach do not have a free-energy axis, which makes it challenging to incorporate surface energies and other forms of thermodynamic work into solid-aqueous stability analyses. Adding a free-energy axis for Pourbaix diagrams can also facilitate the evaluation of thermodynamic driving forces between precursors and crystallization products[32], for example, when a $Mn^{2+}$(aq) precursor is under $E$-pH conditions where it is metastable with respect to the nucleation of solid $MnO_2$. To add a free-energy axis to Pourbaix diagrams, here we use a thermodynamic grand potential[33,34], that corresponds to an aqueous ion precursor or metal oxide precipitate in open exchange with a water reservoir at a given pH, redox potential, and dissolved metal ion concentration. Because these are the natural variables of the Pourbaix diagram, we refer to this thermodynamic grand potential as the Pourbaix potential.

We construct the Pourbaix potential, Ψ, by a Legendre transformation of the Gibbs free energy with respect to the oxygen chemical potential, $\mu_O$; the hydrogen chemical potential, $\mu_H$; and redox potential, $E$, under a constraint of water-oxygen equilibrium. Details of this Legendre transformation can be found in the Methods section, and the final expression for the Pourbaix free energy, normalized by number of metal atoms, is expressed as:

$$\overline{\Psi} = \frac{1}{N_{Mn}}\left(\left(G - N_O\mu_{H_2O}\right) - RT\cdot\ln(10)\cdot(2N_O - N_H)pH \quad (2)\right.$$
$$\left. -(2N_O - N_H + Q)E\right)$$

Where $G$ is the molar Gibbs formation free energy; $N_M$, $N_O$, and

**Table 1 Thermochemical data for Mn–H₂O solids**

| Phase | $\Delta G^o_f$ at 25 °C (eV/formula) | Source | Surface energy (J/m²) | Source |
|---|---|---|---|---|
| $Mn_3O_4$ | −13.300 | Hem (1983)[20] | 0.96 | Birkner (2012)[45] |
| $\alpha$-$Mn_2O_3$ | −9.132 | Hem (1978)[60] | 1.36 | This work |
| $Mn(OH)_2$ | −6.381 | Hem (1983)[20] | 0.47 | This work |
| $\beta$-MnOOH | −5.670 | This work | 0.53 | This work |
| $\alpha$-MnOOH | −5.763 | Fritsch (1997)[47] | 0.65 | This work |
| $\gamma$-MnOOH | −5.780 | Hem (1983)[20] | 0.84 | This work |
| R-$MnO_2$ | −4.767 | Kitchaev (2015)[39] | 1.33 | This work |
| $\gamma$-$MnO_2$ | −4.787 | Kitchaev (2015)[39] | 1.44 | This work |
| $\beta$-$MnO_2$ | −4.821 | Hem (1983)[20] | 1.55 | This work |

$N_H$ is the composition of the metal oxide/ion; and if the phase is an aqueous ion, the charge, $Q$, normalized by $e^−$ per formula unit. Using the Pourbaix potential, the relative free-energies between metal-containing phases of different compositions can be compared directly, without needing to explicitly evaluate redox reactions.

To apply this grand potential to the Mn–H₂O system, we use the thermochemical dataset shown in Table 1. Although it is possible to compute Gibbs formation energies for both aqueous ions and solid-state phases using ab initio methods[35,36,37], bulk formation free energies for most of the relevant aqueous and solid-state Mn–O–H phases are known experimentally (Supplementary Table 1), which we use in this work. Missing formation energies for the bulk metastable phases $\beta$-MnOOH, R-$MnO_2$, and $\gamma$-$MnO_2$ are supplemented using DFT-SCAN calculations[38], which we have previously shown to give an accurate description of the energetic ordering and enthalpy differences between polymorphs for the manganese oxides[39]. Formation energies for these three metastable compounds are obtained by referencing the free-energy difference of the metastable polymorph against the ground-state phase of the same composition. For Feitknechtite $\beta$-MnOOH, whose structure is not known, we first performed an ab-initio structure prediction by hydrogenating various layered $MnO_2$ phases, as shown in Supplementary Figs. 1 and 2, resulting in a structure with good agreement with experimental XRD patterns (Supplementary Fig. 3), and similar to the recently resolved $\beta$-NiOOH phase (Supplementary Fig. 4)[40]. Further discussion of the $\beta$-MnOOH structure prediction process is detailed in Supplementary Note 1.

To construct a Pourbaix diagram, each phase is represented by a Pourbaix free-energy surface, $\Psi(E,pH)$; as defined in Eq. (2). Example Pourbaix potentials for the manganese oxide system can be found in Supplementary Note 2. The lowest-energy concave envelope formed by the intersection of all competing free-energy surfaces defines the stable phases and their phase boundaries, as shown in Fig. 1a. By projecting these stability regions onto the E–pH plane, the conventional Pourbaix diagram is retrieved, as shown in Fig. 1b. Metastable phases, which do not typically appear on Pourbaix diagrams[31], can be visualized in $\Psi$-E-pH space, as highlighted in Fig. 1c. By computing the intersection of metastable Pourbaix free-energy planes with the planes of the aqueous ions, one can visualize the full aqueous region where a stable or metastable compound is electrochemically supersaturated. In Fig. 1d, we outline phase boundaries for metastable $\beta$-MnOOH, $\gamma$-MnOOH, R-$MnO_2$, and the full supersaturation region for $Mn_3O_4$. Figure 1d shows numerous regions with overlapping metastable phase boundaries, for example, at conditions corresponding to neutral aerated water ($E \sim 0.5$ V, $pH \sim 7$)[41]. Precipitation of manganese oxides under these conditions would tread the bulk stability regions of numerous thermodynamically-competitive phases.

**Nanoscale Pourbaix diagrams**. From bulk energies alone, one cannot distinguish which of the multiple competing metastable phases in Fig. 1d actually precipitates first during crystallization. Calorimetry experiments have shown that metastable oxides can be stabilized at the nanoscale if they have lower surface energy than the equilibrium phases[29]. Because all materials nucleate and grow through the nanoscale, this nanoscale stabilization of metastable oxides should be intimately related to structure selection during materials formation. The Pourbaix potential is a free-energy expression, meaning it can be easily generalized to incorporate surface energies by adding the conjugate variables $\gamma A$ as:

$$\overline{\Psi}(R) = \overline{\Psi}_{Bulk} + \left(\frac{1}{R}\right)\eta\rho\gamma \qquad (3)$$

Where $\gamma$ is the surface energy, $R$ is an 'effective' particle radius—representing the specific surface area in units of Area/Volume, $\eta$ is the unitless shape factor (Area/Volume$^{2/3}$) of the equilibrium particle morphology, and $\rho$ is the volume normalized per mole of metal.

We compute surface energies for all solid manganese oxide and oxyhydroxides using DFT slab calculations, prepared using the efficient creation and convergence scheme we developed in refs. [42,43], and computed using the SCAN metaGGA functional[44]. For each phase, we enumerate the low-index surfaces and their unique terminations, which are used in the Wulff construction to determine their equilibrium particle morphologies, as shown in Fig. 2a. The morphology-averaged surface energies for the $MnO_xH_y$ phases are shown in Table 1. Our DFT-computed surface energies of bixbyite $Mn_2O_3$ and pyrolusite $\beta$-$MnO_2$ are found to be within the error bars of the hydrated surface energies as experimentally measured by Birkner and Navrotsky[45], providing confidence that DFT can calculate accurate surface energies in the manganese oxide system. Further details on surface calculations and surface energy data can be found in Supplementary Note 3.

The size-dependent Pourbaix potential, $\Psi(E, pH, 1/R)$, exists in a four-dimensional thermodynamic space, and can be projected onto the pH–1/R axes at fixed $E$, or the E–1/R axes at fixed pH, to construct size-dependent Pourbaix diagrams. To highlight the energetic competition between manganese oxide phases at the nanoscale, Fig. 2 shows two example Mn–H₂O nanoscale-Pourbaix diagrams; one varying in redox potential at a fixed pH = 11, and one varying in pH at a fixed redox potential of $E = +0.5$ V.

Surface energy contributions drive three major effects at the nanoscale. First, because surface energy is always positive, the stability regions of all solid phases shrink relative to the aqueous ions as particle size is decreased. Phases that can be stabilized at

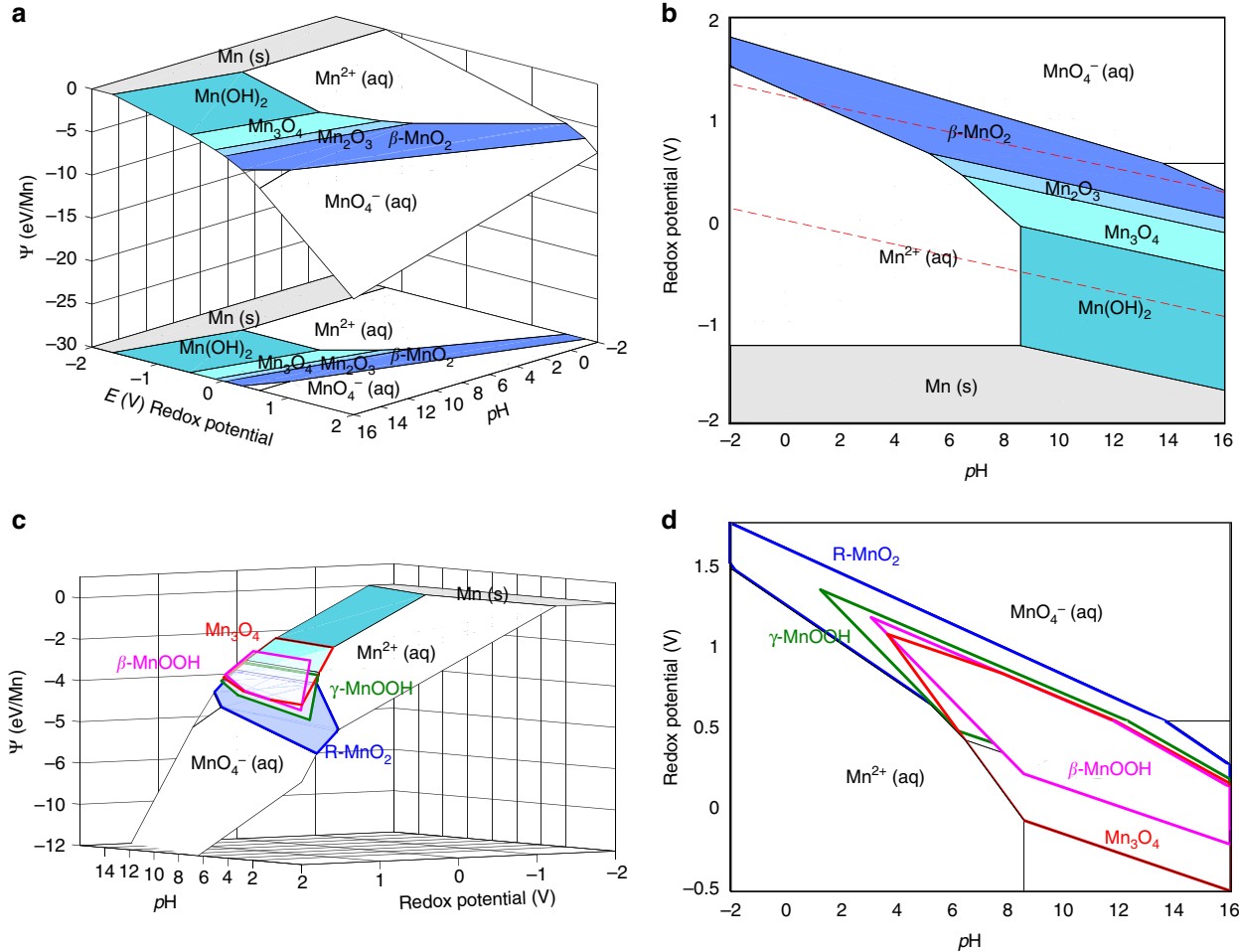

**Fig. 1** Construction of Mn–H$_2$O Pourbaix diagrams using the Pourbaix thermodynamic potential. **a** Concave lowest-energy envelope of Pourbaix free-enesrgy planes in system at [Mn] = 10$^{-2}$ M and 25 °C. **b** Stability regions of equilibrium Mn–O–H phases as projected onto $E$-pH axis. Red dashed lines correspond to redox stability window of water. **c** Pourbaix free-energy planes of metastable β-MnOOH, γ-MnOOH, R-MnO$_2$, and full aqueous stability region of Mn$_3$O$_4$. **d** Projection of aqueous metastability regions onto $E$-pH axis

small particle sizes will require smaller fluctuations during nucleation to grow beyond the critical nucleation radius. These nanoscale Pourbaix diagrams can therefore be used as proxies to estimate $E$-pH conditions where nucleation and crystal growth initiates most readily. Second, as measured by Birkner and Navrotsky, the surface energies of the stable manganese oxide solids are ordered $\gamma_{Mn_3O_4} < \gamma_{Mn_3O_3} < \gamma_{\beta-MnO_2}$[45] As particle size radius is reduced, our nanoscale Pourbaix diagrams show an enlargement of the Mn$_3$O$_4$ stability field and reduction of the β-MnO$_2$ stability field, effectively corresponding to shifts in oxidation-reduction equilibria at the nanoscale[30]. Previously, the precipitation of non-equilibrium oxides have been attributed to kinetic limitations in the transport of oxygen or electron reactants[16,20,46]. Our size-dependent Pourbaix diagrams show that formation of non-equilibrium phases can also have thermodynamic origins, whereby nanoscale shifts in redox equilibria influence the surface work involved in forming nuclei, in turn modifying the kinetics of nucleation.

Third, as shown in Table 1, we calculate the surface energies of MnOOH polymorphs to be ordered $\gamma_{\beta-MnOOH} < \gamma_{\alpha-MnOOH} < \gamma_{\gamma-MnOOH}$, and for the MnO$_2$ polymorphs $\gamma_{R-MnO_2} < \gamma_{\gamma-MnO_2} < \gamma_{\beta-MnO_2}$. The bulk energies of these phases are ordered in the opposite direction, which lead to the aforementioned polymorph stability crossovers at the nanoscale. These inverse relationships between bulk stability and

surface energy within each composition may originate from the fact that a metastable structure has less cohesive energy than a stable phase, which implies a lower energy of cleavage —e.g., a lower surface energy. Interestingly, the MnOOH phases are all calculated to have significantly lower surface energies than the MnO$_2$ phases, even in isostructural manganese oxide frameworks, such as between γ-MnOOH/β-MnO$_2$, and α-MnOOH/R-MnO$_2$[47]. This can be rationalized by the H atoms on the cleaved MnOOH surfaces passivating what would otherwise be broken bonds on the bare isostructural MnO$_2$ surfaces. The low surface energies of these MnOOH phases rationalize why MnOOH compounds readily precipitate in solution, even though they are measured by calorimetry to be thermodynamically unstable on the bulk Pourbaix diagrams, as shown in Fig. 1b.

**Crystallization pathways in redox-active systems.** Multistage crystallization initiates from a metastable precursor, and cascades down in free-energy to the equilibrium phase by a series of phase transformations. To compute a crystallization pathway using the Gibbs free energy, one would evaluate the most favorable series of downhill reactions through a complicated redox reaction network involving the exchange of hydrogen, oxygen, and electrons[48]. However, under the Pourbaix grand potential, all redox reactions

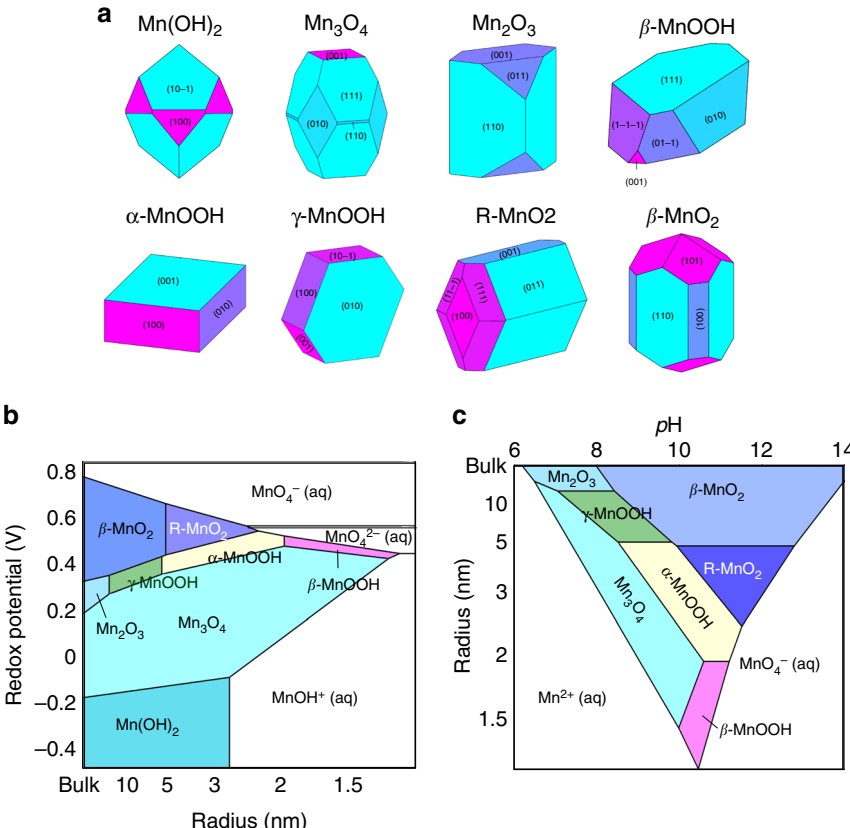

**Fig. 2** Surface energy contributions to Mn–H$_2$O phase equilibria at the nanoscale. **a** DFT-computed Wulff constructions of the equilibrium particle morphologies of size-stabilizable manganese oxide phases. Detailed surface energy data can be found in Supplementary Note 3, and Supplementary Tables 2-11, and Supplementary Figs. 5-14. Wulff constructions are color-coded by the relative surface energies of each facet; blue indicates lower relative surface energy, magenta indicates higher. **b** Size-dependent Pourbaix diagrams at [Mn] $= 10^{-2}$ M and 25 °C, with varying redox potential at fixed pH $= 11$, and with **c** varying pH, at fixed redox potential $E = 0.5$ V

at a given aqueous $E$, pH and ion concentration are evaluated implicitly, meaning one can directly construct a one-dimensional free energy ordering between phases of varying composition. By using the Pourbaix free energy as the electrochemical supersaturation in classical nucleation theory, we arrive at a preliminary theoretical framework to evaluate transition metal oxide crystallization pathways in aqueous solution. Using this framework, we demonstrate that subtle variations in $E$ and pH can modify which metastable phases occur on the multistage oxidation pathway of Mn$^{2+}$(aq), even when these reactions occur within the same $\beta$-MnO$_2$ stability region of the Pourbaix diagram.

Varying $E$ and pH within a phase stability region on the Pourbaix diagram does not change the equilibrium phase, but it can shift the metastable energy landscape, altering the thermodynamic driving forces between precursors, intermediates and products. Figure 3 shows a $\Delta\Psi_{MnO2}$–pH slice of the free energy planes from Fig. 1a, at a fixed $E = +0.5$ V, which is representative of the redox potential in aerated water[41]. The dashed lines in Fig. 3 show that $\Psi_{Mn2+}$ increases linearly with ln[Mn$^{2+}$] concentration, consistent with our traditional intuition regarding supersaturation. At the redox potential shown in Fig. 3, higher [Mn$^{2+}$] activity enlarges the stability region for Mn$_2$O$_3$, and also increases the supersaturation to $\beta$-MnO$_2$.

However, Fig. 3 also shows that the Mn$^{2+}$(aq) supersaturation is strongly dependent on pH, which is not obvious from a Pourbaix diagram or from ideal solution models. Chemically, a high pH signifies a high concentration of OH$^-$ ions, which provides a thermodynamic driving force for the oxidation of Mn$^{2+}$(aq). This oxidation strength can be directly read off of Fig. 3, by

the $\Delta\Psi$ between for example, a metastable Mn$^{2+}$(aq) ion and a metastable MnOOH phase, which increases with increasing pH. Although not shown in Fig. 3, a similar oxidation driving force is achievable through positive redox potentials. Not only do $E$ and pH affect the Pourbaix free-energies of aqueous ions, Equation 2 shows that they also affect the relative $\Delta\Psi$ between solids of different $N_O$ and $N_H$–in other words, manganese oxides of different composition. In Fig. 3b, the influence of pH on the free energy differences between the solids Mn$_3$O$_4$, Mn$_2$O$_3$, and the polymorphs of MnOOH and MnO$_2$ is visualized. Altogether, Fig. 3 reveals a dynamic metastable free-energy landscape over varying electrochemical conditions; a fact that is not readily apparent in traditional Pourbaix diagrams.

This shifting free-energy landscape can lead to variations in multistage crystallization pathways, even when starting with the same precursor and ending with the same equilibrium phase. In Fig. 4, we construct Pourbaix free-energy orderings of manganese (oxyhydr)oxide phases at three conditions within the equilibrium $\beta$-MnO$_2$ stability window; at pH $= 8$, 9.5, and 11, with $E = +0.5$ V and [Mn$^{2+}$] $= 10^{-2}$ M. We compute the oxidation pathway of Mn$^{2+}$(aq) under these conditions, using the procedure we derived in ref. [25], described here briefly:

Starting from the metastable [Mn$^{2+}$] precursor, we compute the steady-state nucleation rate, $J$, to all lower free-energy phases, by

$$J \propto \exp\left(-\frac{(\eta\gamma)^3}{(\Delta\Psi)^2}\right) \qquad (7)$$

where the traditional term for supersaturation, $\Delta G = -RT \ln \sigma$, is

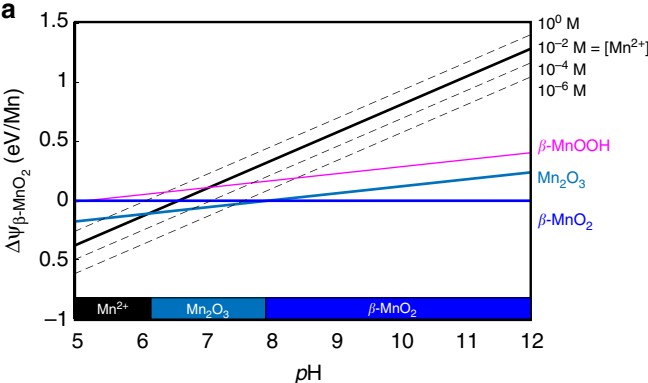

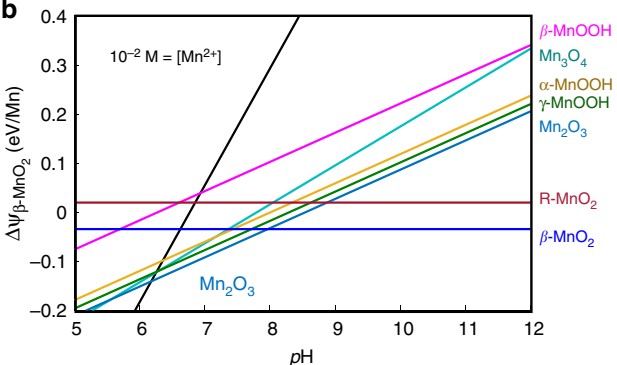

**Fig. 3** Supersaturation of manganese oxides in solution. Shown for various pH at $E = 0.5$ V. **a** Pourbaix free-energy differences between the $Mn^{2+}$(aq) ion and the $\beta$-$MnO_2$ phase, at varying $[Mn^{2+}]$ concentrations (dashed lines). The projection of the lowest free-energy phases onto the pH axis marks the phase stability regions on the Pourbaix diagram. **b** Pourbaix free-energy differences between various solid phases and the $\beta$-$MnO_2$ phase, magnified in the energy axis

replaced by the Pourbaix potential, $\Delta\Psi$. As shown in the nanoscale Pourbaix diagrams, many of the metastable manganese oxides and oxyhydroxides have lower surface energies than the equilibrium phases. Because the nucleation barrier scales with $\gamma^3$, a low surface energy can compensate for a smaller thermodynamic driving force, $\Delta\Psi$, resulting in faster nucleation rates. The induction time of a nucleation event is proportional to the inverse of the nucleation rate, $\tau \sim 1/J$, meaning a fast-nucleating metastable phase can grow and consume $Mn^{2+}$ ions prior to the induction of more-stable phases. Even if the metastable phase completes crystal growth by consuming $Mn^{2+}$ to equilibrium, $\Psi_{Mn^{2+}}$ is still supersaturated with respect to the more stable phases. The next lowest-barrier phase nucleates, and this process repeats by dissolution-reprecipitation in a recursive, energetically-cascading series of metastable stages down to the equilibrium phase, $\beta$-$MnO_2$[25].

Figure 4a shows the computed $Mn^{2+}$ oxidation pathways under the three considered pH conditions. Near the $\beta$-$MnO_2$ phase boundary at pH = 8, thermodynamic driving forces are small; meaning nucleation barriers are high, induction times are long, and metastable intermediates will be long-lived. From the balance of surface energies and bulk driving forces, we compute a crystallization pathway of $Mn^{2+}$(aq) → Hausmannite $Mn_3O_4$ → Groutite $\gamma$-MnOOH → Bixbyite $Mn_2O_3$ → Pyrolusite $\beta$-$MnO_2$. This progression is qualitatively comparable to the observed crystallization pathways in refs. [16,20], which proceeded in freshwater at 25 °C over the course of eight months. At higher pH, the increasingly stratified $\Delta\Psi$ between metastable phases

decreases the induction lifetimes of the transient metastable phases, and qualitatively changes the crystallization pathways; forming $\beta$-MnOOH and bypassing $Mn_2O_3$ at pH = 9.5; and at pH = 11, forming $\beta$-MnOOH and Ramsdellite $MnO_2$.

Altogether, these results indicate that hidden above each equilibrium Pourbaix stability region is a complex metastable energy landscape, where free-energy differences between competing phases vary continuously with $E$ and pH. As summarized in Fig. 4b, these variations can redirect a crystallization pathway down through different metastable phases, even when crystallization initiates from the same precursor and ends within the same stability region of the Pourbaix diagram. These findings rationalize why non-equilibrium crystallization pathways are so sensitive to solution conditions, and highlight the limitations of using an equilibrium phase diagram to guide materials synthesis[49].

## Discussion

In this work, we constructed a unified theoretical framework that bridges aqueous electrochemical Pourbaix diagrams, nanoscale crossovers in phase stability, and classical nucleation theory. This combined analysis distinguishes the subtle roles of thermodynamics and nucleation kinetics during the multistage crystallization of redox-active transition metal oxides. In the manganese oxides, we find that surface energy, redox potential and pH all operate on a similar energy scale of manganese oxide metastability[50], and that all three are important thermodynamic drivers of structure-selection during manganese oxide precipitation. However, the initial precipitation of a non-equilibrium phase can consume much of the reaction driving force, leading to long-lived metastable intermediates and slow nucleation kinetics to the ensuing lower free-energy phases. This delicate balance between thermodynamics and kinetics underlies the complicated dynamics of multistage crystallization, where even in the same phase stability region of a Pourbaix diagram, subtle variations in solution parameters can change which metastable phases a non-equilibrium crystallization pathway passes through. Mapping this metastable energy landscape provides a combined thermodynamic and kinetic foundation for guiding the targeted synthesis of functional metal oxides.

To conclude, we discuss opportunities to achieve more quantitative predictions from this theoretical framework. Is it known that applied pH and redox potential can influence the interfacial structure and adsorption dynamics of the electrochemical double layer[27,51,52]. A more quantitative understanding of the thermochemistry of the electrochemical double layer, especially as a function of surface chemistry and electronic structure, will provide an additional handle to engineer crystallization pathways in solution. Additionally, solid-solid transformations, for example, by H-ion or Mn-ion migration within an oxygen sublattice[53,54], could be a competing structure transformation mechanism to dissolution-reprecipitation[55]. The energy barriers of ion-diffusion (units of meV/atomic hop) and crystal nucleation (units of meV/nucleus) have different units, and therefore the relative kinetics between these two competing mechanisms cannot be directly compared. A theoretical framework that can treat dissolution-reprecipitation and solid-state transformations on equal footing would enable a more quantitative comparison of the kinetics between diffusive and displacive phase transformations. Finally, we emphasize the need for more deliberate measurements of the redox potential during hydrothermal synthesis, which is an important measure of the oxygen fugacity in water. The redox potential can vary with the concentration of dissolved $O_2$ gas, the presence of oxidizing or reducing counterions, and even the

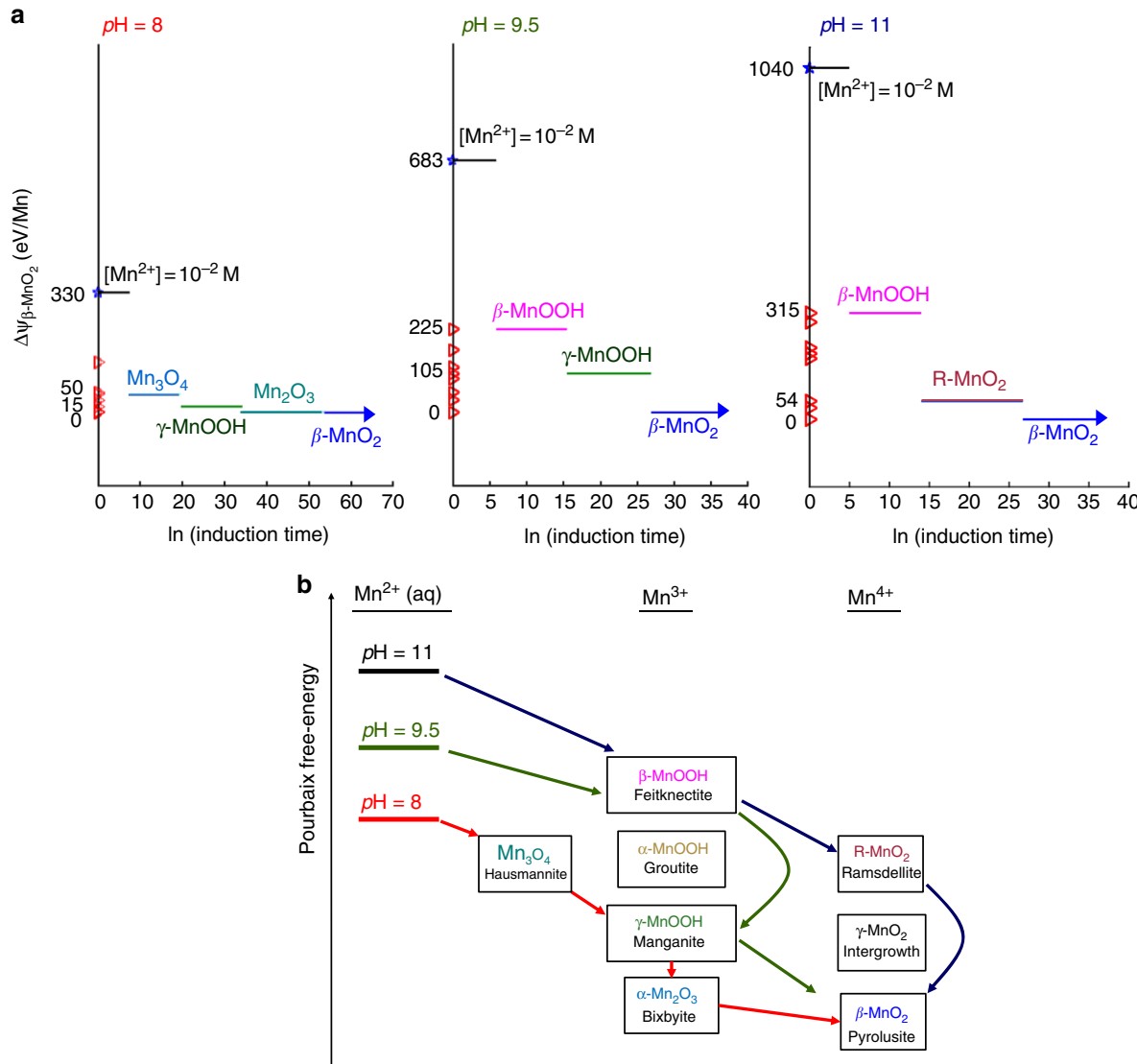

**Fig. 4** Oxidation pathways of $Mn^{2+}_{(aq)}$ at three pH conditions in the $\beta$-MnO2 stability region. **a** Induction time diagrams, with phases ordered vertically by the $\Delta\Psi$ of a phase above $\beta$-MnO2. From the $Mn^{2+}$ precursor, the lowest-barrier phase nucleates and grows, persisting until the induction of the next more-stable phase. The process then repeats. **b** Schematic diagram of the three crystallization pathways as a cascade through the metastable energy landscape; energy axis not drawn to scale

relative humidity in air during sample preparation. Careful potentiometric measurements prior to hydrothermal synthesis is necessary to calibrate the thermodynamic redox conditions under which a precipitation reaction is initiated.

## Methods

**A Legendre transform approach to the Pourbaix potential**. The Pourbaix potential, $\Psi$, is a thermodynamic grand potential, and is constructed by a Legendre transformation of the Gibbs free energy with respect to the oxygen chemical potential, $\mu_O$; the hydrogen chemical potential, $\mu_H$; and redox potential, $E$, under a constraint of water-oxygen equilibrium:

$$\Psi = G - \frac{\partial U}{\partial N_H} \cdot N_H - \frac{\partial U}{\partial Q} \cdot Q - \frac{\partial U}{\partial N_O} \cdot N_O \qquad (8)$$

The partial derivative with respect to charge, $Q$, is the electrical potential, $E$; and the partial derivative with respect to the number of oxygen atoms is the chemical potential of oxygen, $\mu_O$. In solution, the derivative with respect to the number of hydrogen atoms is the chemical potential of a proton $\mu_H+$ at the reference potential ($E = 0$ V vs. SHE) minus the electric work $E$ required to bring the hydrogen atom

into the phase at $E$. The thermodynamic potential can thus be expressed as

$$\Psi = G - (\mu_{H^+} - E) \cdot N_H - E \cdot Q - \mu_O \cdot N_O \qquad (9)$$

In an aqueous system, $\mu_O$ and $\mu_H+$ are constrained by the water-oxygen equilibrium

$$H_2O \leftrightarrow 2 \cdot H^+ + 1/2 \cdot O_2 + 2e^- \qquad (10)$$

which yields

$$\mu_O = \mu_{H_2O} - 2 \cdot \mu_{H^+} + 2E \qquad (11)$$

The number of metal atoms are conserved in the phase transformations between metal oxides with different compositions. Thus, $\Psi$ should be normalized by the number of metal atoms. By substituting Eq. M.4 into Eq. M.2 and normalizing by number of metal atoms, we obtain the Pourbaix potential:

$$\overline{\Psi} = \frac{1}{N_M}\left((G - N_O\mu_{H_2O}) + (2N_O - N_H)\mu_{H^+} - (2N_O - N_H + Q)E\right) \qquad (12)$$

The molar Gibbs free energy of a phase, $G$, is its chemical potential, $\mu_i = \mu_i^o + RT\ln[a_i]$, where $\mu_i^o$ is given by the standard-state Gibbs formation free-energy, $\Delta G^o_f$. An ideal solid with no defects has an activity of one, and so the $RT\ln[a_i]$ term is zero,

but the chemical potential of metal ions ideally scales with the natural log of the metal ion concentration in solution. The chemical potential of protons can be transformed to pH by the relationship $\mu_{H^+} = -RT \cdot \ln(10) \cdot$ pH.

The Pourbaix potential can be further extended to include the thermodynamic effect of intercalating aqueous impurity ions, as shown in ref. [13,18] for intercalating $Li^+$, $K^+$, $Na^+$, $Mg^{2+}$, and $Ca^{2+}$ into various $MnO_2$ polymorphic frameworks, by performing another Legendre Transformation on the bulk Gibbs free energy, replacing $G$ in Eq. (12) by $\Phi = G - \mu_M N_M$, where $M$ is the relevant aqueous metal ion specie.

**Bulk formation free energies.** As shown in Table 1, bulk formation energies are obtained from experimental sources for all phases except $\beta$-MnOOH, $\gamma$-$MnO_2$, and R-$MnO_2$. For these three metastable compounds, experimental bulk formation free energies do not exist or are not reliable. Formation free energies of these three compounds are obtained from DFT, referenced against their equilibrium polymorphs $\gamma$-MnOOH and $\beta$-$MnO_2$. For example,

$$\Delta G^\circ_{f,\beta-MnOOH} = \Delta G^\circ_{f,\gamma-MnOOH} + \left( E^{DFT}_{\beta-MnOOH} - E^{DFT}_{\gamma-MnOOH} \right) \quad (13)$$

Here we assume that the Gibbs free-energy differences between the stable and metastable polymorphs at 298 K are dominated by enthalpy differences. This is a reasonable assumption at room temperatures, where the TΔS term between polymorphic oxides is generally small. The structure of $\beta$-MnOOH has not been previously reported, so we perform an ab initio structure prediction to resolve this crystal structure, as discussed in Supplementary Note 1.

**DFT calculations.** DFT calculations were performed using the Vienna Ab-Initio Software Package (VASP)[56]. We used the projector augmented wave (PAW)[57] method with the strongly-constrained and appropriately-normed (SCAN)[58] metaGGA (generalized-gradient approximation) functional. Plane-wave basis cutoff energies were set at 520 eV for all calculations. Brillouin Zones were sampled using Gaussian smearing, with at least 1000 $k$-points per reciprocal atom for bulk unit cells, and at least 700 $k$-points per reciprocal atom for surface slabs. Atoms were initially relaxed until forces were 1E-6 eV/Å. All structure preparations were performed using the Python Materials Genomic (Pymatgen) package[59].

**DFT surface calculations.** Surface energies of $MnO_xH_y$ phases are calculated in density functional theory, using surface slabs generated using the efficient creation and convergence scheme developed by Sun and Ceder[42]. For each conventional bulk unit cell, low-index surfaces are enumerated[43], and surface energies are calculated within the SCAN metaGGA functional. Surface energy calculations were performed on surface slabs at least 15 Å thick and with 16 Å vacuum. Further details about surface calculations and surface energy data can be found in Supplementary Note 3.

## Data availability
All data necessary to support the findings of this study is available in the manuscript or in the Supplementary Information. Further data and methods can be made available from the authors upon request.

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

## Acknowledgements

This work was supported by the US Department of Energy, Office of Science, Basic Energy Sciences, under Contract no. UGA-0-41029-16/ER392000 as a part of the DOE Energy Frontier Research Center "Center for Next Generation of Materials by Design: Incorporating Metastability." D.Kramer acknowledges support from DAAD (D/06/47916) and EPSRC (EP/R042063/1, EP/R023662/1, and EP/R021295/1). WS used computing resources at the Argonne National Laboratory Center for Nanoscale Materials, an Office of Science User Facility, which was supported by the U. S. Department of Energy, Office of Science, Office of Basic Energy Sciences, under Contract No. DE-AC02-06CH11357. This research also used resources of the Center for Functional Nanomaterials, which is a U.S. DOE Office of Science Facility, at Brookhaven National Laboratory under Contract No. DE-SC0012704. W.S. thanks SYC for valuable insights.

## Author contributions

W.S. and G.C. designed the study. W.S. and D.A.K. performed computational experiments. D.K. formulated the Legendre transform approach to the Pourbaix potential. W.S. analyzed the data, made the figures and prepared the manuscript. All authors discussed the results and commented on the manuscript.

## Additional information

**Competing interest:** The authors declare no competing interests.

