## [Peer Review File · Nature Communications]

Reviewers' comments:

Reviewer #3 (Remarks to the Author):

I found the authors had satisfactorily address my questions from round 1) where I was reviewer 3.

Except: 1) Bixbyite and Hausmanite are not water oxidation catalysts- they are precursors. This should be fixed.

2) I would would have liked some discussion about redox potential

(Redox potential and oxygen fugacity) just because it is a common measure.

Overall while I find this paper still on the specialised end of the scale - it is a lovely story and recommend its publication in *Nature Communications*.

Reviewer #4 (Remarks to the Author):

I have been asked to serve as an adjudicatory reviewer for this submission, which has previously been reviewed by three experts in the field. As such, I will jump immediately to my analysis of the relevance and impact of the work, comparing and contrasting my own conclusions to those of the reviewers and to what the authors have written in their response.

Reviewer 1 clearly had the most extensive comments on the manuscript, and I will therefore focus my remarks on his/her comments and the corresponding responses from the authors. To summarize my conclusions at the outset, I agree with some, although not all, of the reviewer's points, and with some, although not all, of the authors' responses. I also agree with the original decision against publication in *Nature Materials*, while I think that publication in *Nature Communications* might be

possible provided that the authors make very significant changes to the structure and wording of the article.

First, I fully agree with Reviewer 1 that the authors have tremendously exaggerated the significance of their “Pourbaix Potential.” I do not dispute the usefulness of this approach, but it is not at all new, and the formalism has been used countless times in the literature. It is therefore entirely inappropriate for the authors to claim that this is a new development on their part, as they have repeatedly done in the manuscript, and this problem must absolutely be corrected. It is true, as the authors state, that calculating and depicting Pourbaix diagrams by investigation of redox reactions to identify phase boundaries is common practice in the materials community, to which the authors belong (it is perhaps for this reason that they appear to be unaware of the relevant literature). However, plotting grand canonical free energies (of which the Pourbaix Potential is just one particular example) against chemical potentials to identify both stable and metastable phases at given conditions of chemical potential has been done for at least 15 years in the theoretical heterogeneous catalysis and theoretical electrocatalysis communities, dating back to early work of Scheffler (PRL’s in 2003 and 2004, and related references), in which the chemical potentials were subsequently converted to a temperature and pressure scale, much as the present authors convert chemical potentials to a potential and pH scale. The first specific discussion of this approach in the context of electrocatalysis was given by Hansen and Norskov (PCCP 2008, p. 3722, 10.1039/b803956a), who clearly showed the usefulness of calculating grand canonical free energies (which they expressed in terms of electrode potential at fixed pH, using a simple transformation between chemical potential and voltage). The formalism has also been used many, many times in the intervening years, with the two references frequently mentioned by reviewer 1 (Nature Energy 2, 17070 (2017), J. Phys. Chem. C, 2015, 119 (32), pp 18177, and J. Phys. Chem. C, 2017, 121 (18), pp 9782).

The authors’ response to these comments from reviewer 1 is highly inadequate (they simply state that they did not claim that their construction of metastable Pourbaix diagrams is novel, but in fact, they have strongly implied this throughout the document and have claimed multiple times that they “derived” this potential). I would say that the first seven pages of the manuscript are devoted to deriving and developing a formalism that, as stated above, is perfectly well known. Indeed, it is completely inappropriate to claim that they are developing a derivation of any kind. The discussion on p.11 is similarly overstated (“a major advantage of the Pourbaix potential...”). To be published anywhere, the authors must condense the first seven pages to about 1-2 pages, wherein they should focus their discussion on the need for incorporation of nanoscale features into Pourbaix diagrams, and they should simply refer readers to the enormous literature on use of grand canonical potentials in calculated phase diagrams for the rest. Simply stating eq. 2 would be sufficient, with no derivations and no flashy figures.

Having said the above, I do agree with the authors that adding a nanoscale component to Pourbaix diagram analysis is interesting and has not, to my knowledge, been done in the manner that the authors present. The combination of equations (3) and (7), together with the discussion on p.13 and the results in Figure 5, does indeed provide a compact, useful, and easy-to-understand strategy for

qualitatively estimating how dissolution-precipitation-mediated crystallization pathways may change as a function of pH and potential. This is the new part of the work, and as stated above, this should constitute the vast majority of a revised or resubmitted manuscript.

One additional technical comment concerns the statement on p. 10 that “These nanoscale shifts in redox equilibria offer an alternative mechanism to rationalize the precipitation of non-equilibrium oxide compositions, the formation of which has thus far been attributed primarily to kinetic limitations in electron or oxygen transport.” This statement is not wrong, but it should be qualified, since the authors’ own arguments and conclusions later in the manuscript clearly show that the kinetics of dissolution/precipitation will be very relevant to determining which (if any) metastable phases along the pathway will ever actually be observed. Clearly, the phenomenon does not depend solely on local redox equilibria.

“Non-equilibrium crystallization pathways of manganese oxides in aqueous solution”

Reviewer Response to Nature Communications NCOMMS-18-16028-T

September 22nd, 2018

Changes in the manuscript have been highlighted in Blue for Reviewer #3, and Yellow for Reviewer #4.

Reviewers' comments:

Reviewer #3 (Remarks to the Author):

I found the authors had satisfactorily address my questions from round 1) where I was reviewer 3.

Except: 1) Bixbyite and Hausmanite are not water oxidation catalysts- they are precursors. This should be fixed.

We thank Reviewer #3 for pointing this out, we have made this correction, highlighted in blue in the first paragraph on Page 2.

2) I would have liked some discussion about redox potential

(Redox potential and oxygen fugacity) just because it is a common measure.

By Reviewer #3's recommendation, we have added more discussion on redox potential and oxygen fugacity and its importance during hydrothermal synthesis, as highlighted in blue in the conclusion paragraph.

Overall while I find this paper still on the specialised end of the scale - it is a lovely story and recommend its publication in Nature Communications.

We very much thank Reviewer #3's positive assessment of our manuscript.

Reviewer #4 (Remarks to the Author):

I have been asked to serve as an adjudicatory reviewer for this submission, which has previously been reviewed by three experts in the field. As such, I will jump immediately to my analysis of the relevance and impact of the work, comparing and contrasting my own conclusions to those of the reviewers and to what the authors have written in their response.

Reviewer 1 clearly had the most extensive comments on the manuscript, and I will therefore focus my remarks on his/her comments and the corresponding responses from the authors. To summarize my conclusions at the outset, I agree with some, although not all, of the reviewer's points, and with some, although not all, of the authors' responses. I also agree with the original decision against publication in Nature Materials, while I think that publication in Nature Communications might be possible provided that the authors make very significant changes to the structure and wording of the article.

We are grateful to Reviewer #4 for serving as an adjudicatory reviewer for this work, and for the careful work that Reviewer #4 has put into evaluating our manuscript.

First, I fully agree with Reviewer 1 that the authors have tremendously exaggerated the significance of their "Pourbaix Potential." I do not dispute the usefulness of this approach, but it is not at all new, and the formalism has been used countless times in the literature. It is therefore entirely inappropriate for the authors to claim that this is a new development on their part, as they have repeatedly done in the manuscript, and this problem must absolutely be corrected. It is true, as the authors state, that calculating and depicting Pourbaix diagrams by investigation of redox reactions to identify phase boundaries is common practice in the materials community, to which the authors belong (it is perhaps for this reason that they appear to be unaware of the relevant literature). However, plotting grand canonical free energies (of which the Pourbaix Potential is just one particular example) against chemical potentials to identify both stable and metastable phases at given conditions of chemical potential has been done for at least 15 years in the theoretical heterogeneous catalysis and theoretical electrocatalysis communities, dating back to early work of Scheffler (PRL's in 2003 and 2004, and related references), in which the chemical potentials were subsequently converted to a temperature and pressure scale, much as the present authors convert chemical potentials to a potential and pH scale. The first specific discussion of this approach in the context of electrocatalysis was given by Hansen and Norskov (PCCP 2008, p. 3722, 10.1039/b803956a), who clearly showed the usefulness of calculating grand canonical free energies (which they expressed in terms of electrode potential at fixed pH, using a simple transformation between chemical potential and voltage). The formalism has also been used many, many times in the intervening years, with the two references frequently mentioned by reviewer 1 (Nature Energy 2, 17070 (2017), J. Phys. Chem. C, 2015, 119 (32), pp 18177, and J. Phys. Chem. C, 2017, 121 (18), pp 9782).

Reviewer #1 and #4 have both made significant comments on the previous use of grand potentials in the computational catalysis communities, citing references by Scheffler, Norskov, and others. We do agree that grand potentials have been around in the computational community for a long time. Accordingly, we have added multiple references (Refs 30-35) to acknowledge these prior contributions.

However, we would argue that grand potentials have not been as widely adopted in the geochemistry, nucleation and crystallization communities, which are the communities that we hope will benefit the most from this paper. Grand potentials are most useful when the product phases of a precipitation reaction can exhibit a wide range of stoichiometries; such as in the transition metal oxides (Mn₃O₄, Mn₂O₃, MnOOH, MnO₂ as was investigated in this work). We invite Reviewer #4 to review some of the current theoretical approaches used to model the aqueous crystallization of transition metal oxides, for example, in:

Hem, John D., and Carol J. Lind. "Nonequilibrium models for predicting forms of precipitated manganese oxides." Geochimica et Cosmochimica Acta 47.11 (1983): 2037-2046.

In this paper, the authors used an Ion Activity Product (IAP) approach to compute Reaction Affinities, which they translate into a crystallization driving forces by $\Delta G = -RT \ln(IAP/K_{sp})$. One limitation of this theoretical approach is that there can be multiple possible redox reactions in water. For example, in just the aqueous reaction from Mn²⁺(aq) to solid MnO₂, one could write the following redox reactions:

All of these reactions are valid, but they each take into account different aspects of the aqueous chemistry (pH, redox potential, etc). One could write additional constraining equations to relate H_2O , O_2 , e^{-} , H^{+} and OH^{-} , but this results in a convoluted reaction network for evaluating the thermodynamics of precipitation reactions in redox-active systems.

For a more recent example, consider:

Baumgartner, Jens, et al. "Nucleation and growth of magnetite from solution." *Nature Materials* 12.4 (2013): 310.

In this work, the authors compared the relative nucleation rates between ferrihydrite (FeOOH) and magnetite (Fe_3O_4) starting from aqueous iron precursors. However, they used the Gibbs Free Energy in calculating the nucleation barrier, resulting in a *formally incorrect* approach to compare driving forces from an aqueous Fe(III) precursor to two product phases, Fe_3O_4 and FeOOH , which have different compositions. The analyses by these authors would have been made valid if they were using a thermodynamic grand potential, instead of the Gibbs free energy.

We highlight these two highly-cited examples from the literature to illustrate how the nucleation and crystallization communities are in urgent need of a concise and formally-correct equation to properly evaluate the nucleation driving force for the precipitation of transition metal oxides. It is our belief that the Pourbaix potential is precisely what is needed to accomplish this task.

In conclusion, we are not trying to diminish what has been achieved by grand potentials in the catalysis communities, and in the revised manuscript we have rewritten the text to properly acknowledge those previous contributions. Rather, our goal in this work was to cross-pollinate the theoretical advances that were made in the computational catalysis fields to address urgent needs in the nucleation and crystallization communities.

The authors' response to these comments from reviewer 1 is highly inadequate (they simply state that they did not claim that their construction of metastable Pourbaix diagrams is novel, but in fact, they have strongly implied this throughout the document and have claimed multiple times that they "derived" this potential). I would say that the first seven pages of the manuscript are devoted to deriving and developing a formalism that, as stated above, is perfectly well known. Indeed, it is completely inappropriate to claim that they are developing a derivation of any kind. The discussion on p.11 is similarly overstated ("a major advantage of the Pourbaix potential..."). To be published anywhere, the authors must condense the first seven pages to about 1-2 pages, wherein they should focus their discussion on the need for incorporation of nanoscale features into Pourbaix diagrams, and they should simply refer readers to the enormous literature on use of grand canonical potentials in calculated phase diagrams for the rest. Simply stating eq. 2 would be sufficient, with no derivations and no flashy figures.

In our revised manuscript, we have provided a new version based on the recommendations of Reviewer #4. We have rewritten the introduction section to emphasize the size-dependent thermodynamics and nucleation kinetics of redox-active transition metal oxides, significantly diminishing the previous discussion which aimed to set up the 'derivation' of the Pourbaix potential. We then provide a significantly reduced section on the Pourbaix potential, focusing any discussion specifically to set up its use for constructing size-dependent Pourbaix diagrams, and for use as the electrochemical supersaturation (the denominator in the nucleation barrier, relevant to the last

section of the paper). We removed what was previously Figure 1. We have also eliminated any overstatements of us having ‘derived’ the Pourbaix potential from the rest of the manuscript.

This reduces the first 7 pages to 4.5 pages. We note that the abstract and motivation section on MnOxides was previously already 2 pages, and that important thermochemical data also took up about a page of text and table. Shortening it any further would cut into sections that are not about the Pourbaix potential, but rather, about multistage crystallization and the manganese oxide system. We hope that the revised version of the manuscript satisfies the intent of Reviewer #4's request.

As discussed in the previous comment, we do believe that it would be beneficial to future readers, in particular those from outside the catalysis community, if the Legendre Transform approach to the Pourbaix potential could at least remain in the Appendix. As a postscript: We do note that the Pourbaix potential is not as straightforward to construct as a grand potential for μ_{O_2} , which can be done in one-step: $\mu_{\text{O}_2}(T, p_{\text{O}_2}) = \mu_{\text{O}_2}^\circ + RT \ln(p_{\text{O}_2}) - TS_{\text{O}_2}$. In $\Psi(E, \text{pH})$, the Legendre Transform is done with respect to μ_{O} , μ_{H} , and Q ; but then, μ_{O} , μ_{H} and Q is further constrained by the water reduction reaction, $\text{H}_2\text{O} \leftrightarrow 2 \cdot \text{H}^+ + \frac{1}{2} \text{O}_2 + 2 e^-$. It is in this second step that the E and pH dependence arise. In principle, this constrained Legendre transform approach could be generalized to make ‘Pourbaix-like’ diagrams for other redox-active solvents used in solvothermal synthesis, such as ammonia $\text{NH}_3 \leftrightarrow \frac{1}{2} \text{N}_2 + 3\text{H}^+ + 3e^-$, ethanol, ethylene glycol, methylamine, etc. The facile construction of such diagrams would be of great value for synthetic chemists who synthesize solid-state materials via solvothermal methods.

Having said the above, I do agree with the authors that adding a nanoscale component to Pourbaix diagram analysis is interesting and has not, to my knowledge, been done in the manner that the authors present. The combination of equations (3) and (7), together with the discussion on p.13 and the results in Figure 5, does indeed provide a compact, useful, and easy-to-understand strategy for qualitatively estimating how dissolution-reprecipitation-mediated crystallization pathways may change as a function of pH and potential. This is the new part of the work, and as stated above, this should constitute the vast majority of a revised or resubmitted manuscript.

We thank Reviewer #4 for the positive comments on this section. We believe that this discussion does indeed now constitute the vast majority of the revised manuscript.

One additional technical comment concerns the statement on p. 10 that “These nanoscale shifts in redox equilibria offer an alternative mechanism to rationalize the precipitation of non-equilibrium oxide compositions, the formation of which has thus far been attributed primarily to kinetic limitations in electron or oxygen transport.” This statement is not wrong, but it should be qualified, since the authors’ own arguments and conclusions later in the manuscript clearly show that the kinetics of dissolution/reprecipitation will be very relevant to determining which (if any) metastable phases along the pathway will ever actually be observed. Clearly, the phenomenon does not depend solely on local redox equilibria.

We thank Reviewer #4 for this point. We have clarified this section in the revised manuscript, explaining that that non-equilibrium oxide compositions have previously been claimed to originate

from the kinetics of *reactant transport*, whereas here, we show that non-equilibrium oxide compositions can also be due to the kinetics of *nucleation*. These are two qualitatively different mechanisms that unfortunately both fall under the umbrella of 'kinetics', which was confusing in the original version. We have revised this section, as highlighted on the current p.9, to address this potential conceptual pitfall.

We humbly thank Reviewer #3 and Reviewer #4 for their careful reads and valuable suggestions, and believe that the recommended revisions have strengthened our manuscript.

REVIEWERS' COMMENTS:

Reviewer #3 (Remarks to the Author):

I was reviewer 3 and as I noted previously the authors had satisfactorily addressed my previous comments. I still find the paper to be novel particularly in the constructions of nano-sizing in Pourbaix diagrams and worthy of publication in Nature communications. (All be it a little on the specific rather than general interest side)

I found the comments by reviewers 1 and 4 very interesting, and I learnt a lot from reading them and the authors responses to them.

In science what we get out of reading a paper something of a matter of perspective. If I take the comments of reviewer 1 and 4 as read it seems that some of the novel aspects of this paper (which I found very interesting and didn't know) have previously been described elsewhere. Yet I am very familiar with publications on Manganese Oxides and I had never thought about this in the way the authors describe. Therefore I have to agree with the authors even if aspects have been published before (and they should be cited etc) there is still significant value to the catalysis and geochemical communities in this manuscript.

I stand by my original recommendation to publish the work in nature communications- and hopefully the reviewers comments too.